# miR-200c-3p, miR-222-5p, and miR-512-3p Constitute a Biomarker Signature of Sorafenib Effectiveness in Advanced Hepatocellular Carcinoma

**DOI:** 10.3390/cells11172673

**Published:** 2022-08-28

**Authors:** Patricia de la Cruz-Ojeda, Tobias Schmid, Loreto Boix, Manuela Moreno, Víctor Sapena, Juan M. Praena-Fernández, Francisco J. Castell, Juan Manuel Falcón-Pérez, María Reig, Bernhard Brüne, Miguel A. Gómez-Bravo, Álvaro Giráldez, Jordi Bruix, María T. Ferrer, Jordi Muntané

**Affiliations:** 1Institute of Biomedicine of Seville (IBiS), Hospital University “Virgen del Rocío”/CSIC/University of Seville, 41013 Seville, Spain; 2Networked Biomedical Research Center Hepatic and Digestive Diseases (CIBEREHD), 28029 Madrid, Spain; 3Department of Medical Physiology and Biophysics, University of Seville, 41004 Seville, Spain; 4Institute of Biochemistry I, Faculty of Medicine, Goethe-University Frankfurt, 60528 Frankfurt, Germany; 5BCLC Group, Liver Unit, Hospital Clinic, University of Barcelona, IDIBAPS, CIBEREHD, 08036 Barcelona, Spain; 6Department of General Surgery, Hospital University “Virgen del Rocío”/CSIC/University of Seville/IBIS, 41013 Seville, Spain; 7Department of Statistics and Operations Research, University of Granada, 18011 Granada, Spain; 8Department of Radiology, Hospital University “Virgen del Rocío”/CSIC/University of Seville/IBIS, 41013 Seville, Spain; 9Exosomes Lab, CIC bioGUNE, 48160 Derio, Spain; 10German Cancer Consortium (DKTK), Partner Site Frankfurt, 60528 Frankfurt, Germany; 11Frankfurt Cancer Institute, Goethe-University Frankfurt, 60528 Frankfurt, Germany; 12Fraunhofer Institute for Translational Medicine and Pharmacology ITMP, 60528 Frankfurt, Germany; 13Unit for the Clinical Management of Digestive Diseases, Hospital University “Virgen del Rocío”/CSIC/University of Seville/IBIS, 41013 Seville, Spain

**Keywords:** extracellular vesicle, hepatocellular carcinoma, miRNA, liquid biopsy, Sorafenib

## Abstract

Background: Sorafenib constitutes a suitable treatment alternative for patients with advanced hepatocellular carcinoma (HCC) in whom atezolizumab + bevacizumab therapy is contraindicated. The aim of the study was the identification of a miRNA signature in liquid biopsy related to sorafenib response. Methods: miRNAs were profiled in hepatoblastoma HepG2 cells and tested in animal models, extracellular vesicles (EVs), and plasma from HCC patients. Results: Sorafenib altered the expression of 11 miRNAs in HepG2 cells. miR-200c-3p and miR-27a-3p exerted an anti-tumoral activity by decreasing cell migration and invasion, whereas miR-122-5p, miR-148b-3p, miR-194-5p, miR-222-5p, and miR-512-3p exerted pro-tumoral properties by increasing cell proliferation, migration, or invasion, or decreasing apoptosis. Sorafenib induced a change in EVs population with an increased number of larger EVs, and promoted an accumulation of miR-27a-3p, miR-122-5p, miR-148b-3p, miR-193b-3p, miR-194-5p, miR-200c-3p, and miR-375 into exosomes. In HCC patients, circulating miR-200c-3p baseline levels were associated with increased survival, whereas high levels of miR-222-5p and miR-512-3p after 1 month of sorafenib treatment were related to poor prognosis. The RNA sequencing revealed that miR-200c-3p was related to the regulation of cell growth and death, whereas miR-222-5p and miR-512-3p were related to metabolic control. Conclusions: The study showed that Sorafenib regulates a specific miRNA signature in which miR-200c-3p, miR-222-5p, and miR-512-3p bear prognostic value and contribute to treatment response.

## 1. Introduction

Hepatocellular carcinoma (HCC) constitutes the sixth most common type of cancer and is the second leading cause of death by cancer [1]. Treatment allocation and prognosis is currently done according to the Barcelona Clinic Liver Cancer (BCLC) staging system [2]. The landscape of treatments for advanced stage of HCC has been changing due to the approval of atezolizumab plus bevacizumab as first-line treatment [3]. However, patients without preserved liver function (compensated Child–Pugh A), high-risk for bleeding, vascular disorders and arterial hypertension as well as severe autoimmune disorders and prior transplantation are not recommended to receive atezolizumab plus bevacizumab treatment. As a consequence, patients that could not benefit from this therapy might be considered for treatment with Sorafenib [2,4].

Several serological tests such as alpha-fetoprotein (AFP), glycosylated AFP (AFP-L3), des-gamma-carboxy prothrombin (DCP), or dickkopf (DKK1) have been investigated for the detection of HCC [5,6,7]. Although several biomarkers such as angiopoietin 2 and vascular endothelial growth factor have been proposed as predictors of survival in patients with advanced HCC [8], there are no consolidated biomarkers or molecular profiles of treatment efficacy nor HCC prognosis [2,9]. In particular, dysregulation in microRNA (miRNA) expression might be associated with prognosis. Several studies have tried to identify miRNA signatures that could help in deciphering patient outcome in HCC using data repositories. Among them, miR-326, miR-30d-5p, let-7c-5p, miR-5003-3p, miR-760, and miR-7-5p constitute prognostic biomarkers of immune infiltration [10]. Others like miR-139-5p, hsa-miR-326, miR-10b-5p, miR-500a-3p, and miR-592 are representative of necroptosis [11]. Even though these studies provide great knowledge, validation experiments are needed in external patient cohorts [10,11,12,13]. Using a different approach, many other studies have tried to elucidate mechanistic pathways based on the regulation of miRNAs by competing endogenous RNAs such as long-non coding RNAs (lncRNAs) [14] or circular RNAs (circRNAs) [15]. Most of them have been carried out in tumor samples, in vitro or in vivo, insisting on the need of validated data in independent cohorts. Furthermore, liquid biopsy biomarkers such as circulating miRNAs might be useful for clinical making decisions [16]. For instance, high serum miR-518d-5p levels correlated with reduced survival following Sorafenib therapy [17]. In the case of the second-line treatment Regorafenib, miRNAs miR-30a, miR-122, miR-125b, miR-200a, miR-374b, miR-15b, miR-107, miR-320, and miR-645 were associated with increased survival after treatment [18]. 

The aim of the study was to identify miRNA biomarkers that correlate with Sorafenib effectiveness in the treatment of patients with advanced HCC. Therefore, we determined a functional miRNA signature as well as the regulation of EV secretion and their miRNA content in Sorafenib-treated HepG2 cells. The miRNA profile was validated in an in vivo xenograft model and plasma from patients at advanced stages of the disease under Sorafenib treatment. 

## 2. Materials and Methods

### 2.1. Primary Human Hepatocytes, Cell Lines, and Culture Conditions

Human hepatocytes were isolated from liver biopsies (two women and one man, aged 66 ± 3.0 years) submitted to surgical resection for liver tumors after obtaining the patients’ written informed consent [19]. 

The hepatoblastoma HepG2 cell line (HB-8065™, ATCC-LGC Standards, S.L.U., Barcelona, Spain) [20] was cultured in MEM with Earle’s salts with L-glutamine with 10% fetal bovine serum (FBS) (Ref. F7524, Sigma-Aldrich, batch BCBX9154, San Luis, CA, USA), sodium pyruvate (1 mM) (Ref. 11360070, Gibco, Thermo Fisher Scientific, Waltham, MA, USA), non-essential amino acids (Ref. 11140035, Gibco, Thermo Fisher Scientific), penicillin–streptomycin solution (100 U/mL-100 μg/mL) (Ref. 15640055, Gibco, Thermo Fisher Scientific) at 37 °C in a humidified incubator with 5% CO_2_. Cells were cultured at a cell density of 100,000 cells/cm^2^ and Sorafenib (10 µM) (Ref. FS10808; Carbosynth, Berkshire, UK) was added 24 h after plating. 

### 2.2. Patients in Advanced Stage of HCC

Circulating miRNA profiles were tested in two independent cohorts of patients ≥ 18 years old with provided written informed consent after the nature and possible consequences of the studies were explained. Blood was obtained before the initiation of Sorafenib treatment and 1 month after treatment. Appendix A shows the demographic and epidemiological data. A cohort from Hospital University “Virgen del Rocío” (n = 36) (June 2015–August 2021, median follow-up of 10.4 months) was used as the study population and another cohort of patients from BCLC was used as the validation cohort (n = 81) (March 2008–July 2011, median follow-up of 11.4 months). Our study protocol followed the ethical guidelines of the 1975 Declaration of Helsinki and was approved by the local ethical committee (1189-N-20). 

### 2.3. Xenograft Mouse Model

Tumors were generated by subcutaneous injection of 10^7^ HepG2 cells mixed 1:1 with Matrigel^®^ (Ref. 354262, Corning, Corning, NY, USA) in the right flank of male immunocompromised mice (Hsd:Athymic Nude-Foxn1nu) (n = 14) (Charles River, Wilmington, NC, USA). Animals were randomized into the Sorafenib and control groups. Sorafenib (200 mg/kg) was administered daily by oral gavage after the tumors reached a diameter of 0.5 cm. Animals were sacrificed under anesthesia when tumors from the control group reached 1.5 cm. Our experimental design accomplished Animal Research Reporting of In vivo Experiments (ARRIVE) principles for replacement, refinement, and reduction. Animal care was in accordance with the institutional guidelines.

### 2.4. RNA Extraction

Total RNA from the cells, tumors, and EVs was extracted using the miRNeasy Kit (Ref. 217004, Qiagen, Hilden, Germany). Lysis was performed with Qiazol and RNA fractions were bound to the RNeasy Mini spin column, washed, and eluted in RNase-free water. Cellular RNA was then quantified using a NanoDrop™ One/OneC Microvolume UV–Vis Spectrophotometer (Thermo Fisher Scientific). Circulating miRNAs were extracted from 200 µL of plasma using the miRNeasy Serum/Plasma Kit (Ref. 217184, Qiagen). To assess the quality of RNA extraction from the plasma samples and EVs, three spike-in controls were added at the beginning of the extraction (Ref. 339390, Qiagen) (2 fmol UniSp2, 0.02 fmol UniSp4, 0.00002 fmol UniSp5).

### 2.5. miRNA Profiling and Bioinformatic Analysis

miRNAs were profiled by qRT-PCR using the TaqMan^®^ OpenArray^®^ Human miRNA Panel (Ref. 4470187, Life Technologies, Carlsbad, CA, USA) and accessory kits (Applied Biosystems) (GEO accession number GSE201696). Briefly, isolated miRNAs were reverse transcribed into cDNA using the TaqMan MicroRNA Reverse Transcription Kit (Ref. 4366596, Life Technologies) and Megaplex RT Primers Human in a set of predefined pools A and B (Ref. 4444750, Life Technologies). cDNA was preamplified with Megaplex PreAmp Primers of gene-specific Human Pools A and B (Ref. 4391128, Life Technologies). The preamplified product was diluted 40-fold and mixed 1:1 *v*/*v* with the TaqMan OpenArray Real-Time PCR Master Mix (Ref. 4462159, Life Technologies). Aliquots of the mixture were dispensed on a microfluidic OpenArray 384-well sample plate (Ref. 4406947, Life Technologies). TaqMan OpenArray Human MicroRNA Panels were loaded with the OpenArray AccuFill System and the PCR reactions were carried out with QuantStudio™ 12 K Flex OpenArray^®^ Platform (QS12KFlex, Thermo Fisher) following the manufacturer’s instructions. 

Thermo Fisher Cloud Relative Quantification software was used to obtain the qPCR data. The expression levels were calculated using the relative threshold cycle (Crt) method. This assay aims to obtain a total of 758 Crt values for each sample, which includes 754 unique miRNAs, one negative control (ath-miR159a) and three endogenous positive controls (RNU48, RNU44 and U6). Crt values were normalized using the global mean strategy. Therefore, ΔCrt values were calculated as the Crt miRNA-Crt global mean. 

To analyze the microarray data, we included stringent quality control criteria: Crt values < 28, amplification score > 1.15, Cq confidence > 0.8, Cq standard deviation < 0.25, and detectable expression in all samples. miRNAs that did not fulfil these criteria were excluded from the analysis. Relative miRNA expression levels between the control and experimental groups were calculated by using the ΔΔCrt method [21] and exported for further analysis. Fold change values were calculated as 2^−ΔΔCrt^. Differentially expressed miRNAs showing ≥2.0-fold change or <0.5-fold with a *p*-value < 0.05 were used for further experiments.

Target prediction was conducted using the TargetScan and miRDB databases. For TargetScan prediction, only highly conserved predicted targets with a cumulative weighted context score < −0.4 were selected. For miRDB targets, the score was set at 80 points. Enrichment analysis of Gene Ontology (GO) terms and Kyoto Encyclopedia of Genes and Genomes (KEGG) pathways with the Database for Annotation, Visualization, and Integrated Discovery (DAVID) tool [22] allowed us to investigate the biological function of miRNA targets.

### 2.6. Quantitative Real Time PCR (qRT-PCR)

Differentially expressed miRNAs were analyzed by qRT-PCR using the miRCURY LNA™ miRNA PCR System (Qiagen). Mature miRNAs were retrotranscribed using the miRCURY LNA RT Kit (Ref. 339340, Qiagen), which allows for universal cDNA synthesis through the addition of a poly(A) tail to miRNA templates. For the miRNA measurements from cell lysates, 25 ng of RNA was used as the starting material, whereas 7 µL of the eluates were used for cDNA synthesis in the case of conditioned media and extracellular vesicles. In the study involving miRNA measurements in patients, 1.12 µL of RNA eluates were used for cDNA synthesis. qRT-PCR was carried out with miRCURY LNA™ miRNA PCR Assays (Qiagen) in a ViiA™ 7 Real-Time PCR System (Applied Biosystems, Thermo Fisher Scientific). The Ct values were analyzed using the ΔΔCt method and the fold change values were calculated as 2^−ΔΔCrt^ with the Thermo Fisher Cloud Relative Quantification software [21]. U6 and spike-in controls were used as the endogenous and exogenous controls, respectively. 

### 2.7. Transfections with miRNA Mimics and Inhibitors

HepG2 cells (50,000 cells/cm^2^) were transfected with either miRNA mimics or antisense oligonucleotides according to the tested miRNA and their respective transfection controls (miRCURY LNA™ miRNA, Qiagen). Twenty-four hours after plating, cell transfections were carried out using Lipofectamine RNAiMAX reagent according to the manufacturer’s instructions in reduced serum media OptiMEM (Gibco, Thermo Fisher Scientific). miRNA mimics and antisense oligonucleotides were used at effective concentrations (1–50 nM). Transfections were carried out during 6 h to minimize cell toxicity. After, the media were refreshed and experimental procedures concerning proliferation, cell death, migration, and invasion assays were carried out.

### 2.8. Measurement of Cell Death and Proliferation

Caspase-3/7 activity was determined using the Caspase-Glo^®^ 3/7 Assay (Ref. G8091; Promega, Madison, WI, USA). Caspase activity was detected in 25 µg of total protein incubated with 20 µL of assay buffer in white bottom 384-well plates for 45 min at RT in the dark. The cell proliferation rate was measured using a commercial assay based on bromodeoxyuridine (BrdU) incorporation (Ref. 11647229001; Roche Diagnostics, Indianapolis, IN, USA). After BrdU addition, cells were fixed and DNA was denaturalized to allow for antibody binding. The anti-BrdU antibody was incubated for 90 min at RT. Then, a ready-to-use substrate solution was incubated for 5 min until the development of a blue color. 

For caspase measurement, luminescence and absorbance (at 370 nm using 492 nm as the reference wavelength) for proliferation, and absorbance signals were measured using an Infinite 200 PRO Microplate Reader (TECAN, Männedorf, Switzerland).

### 2.9. Wound Healing Assay

Cells were transfected with miRNA mimics and antisense oligonucleotides as described above. Cells were allowed to reach confluence for 48 h post-transfection, starved for 6 h with media containing 2% FBS and then scratches were applied with a pipette tip. Cells were incubated for 24 h with Sorafenib. Image acquisition was conducted at 0 and 24 h with an inverted microscope Olympus IX-71. ImageJ software was used for image analysis.

### 2.10. Invasion Assay

The cell invasiveness properties were measured using the QCM ECMatrix Cell Invasion Assay (Ref. ECM550, Chemicon^®^, Merck Millipore, Burlington, VT, USA). Transfected cells (500,000 cells/mL) in serum free medium with or without sorafenib were added to the inserts. After 24 h of incubation, invasive cells were stained with the staining solution included in the kit and dissolved with 10% acetic acid. Absorbance at 560 nm was measured with an Infinite 200 PRO Microplate Reader (TECAN

### 2.11. EV Isolation 

For EV related procedures, HepG2 cells were cultured in FBS-EVs depleted media. For that, FBS diluted in medium was ultracentrifuged overnight in an Optima L100-XP ultracentrifuge with 70 Ti rotor (Ref. 337922; Beckman Coulter, Brea, CA, USA). 

Experimental EVs were isolated from conditioned media through differential centrifugation. Adhered cells were used for marker assessment by Western blot. Dead cells and debris were removed from the media by centrifuging for 15 min at 300× *g*. The pellet was joined to the cellular fraction. Next, the medium was centrifuged at 2000× *g* for 20 min to obtain a fraction called “Large EVs”. The supernatant was collected and centrifuged again at 20,000× *g* for 30 min to obtain a fraction called “Small EVs”. Eventually, the remaining supernatant was filtered through 0.22 µm and ultracentrifuged at 120,000× *g* for 75 min to obtain “Very Small EVs” in an Optima L100-XP ultracentrifuge with 70 Ti rotor (Beckman Coulter).

### 2.12. Nanoparticle Tracking Analysis

The EV size analysis was carried out by nanoparticle tracking analysis (NTA) in a NanoSight LM10 (Malvern Panalytical, Malvern, UK). Samples obtained at 6 h of treatment were diluted 1:15 in PBS and samples obtained at 24 h of treatment were diluted 1:30 for measurements. Samples were loaded into the NanoSight chamber and recorded as three technical measurement replicates.

### 2.13. Determination of EV Markers

Cellular samples were regularly lysed using a solution including 50 mM 4-(2-hydroxyethyl)piperazine-1-ethanesulfonic acid (HEPES) pH 7.5, 5 mM EDTA, 150 mM NaCl, 1% NP-40 protease inhibitor cocktail (Ref. P8340, Sigma-Aldrich), 1 mM phenylmethylsulfonyl fluoride (PMSF), 1 mM NaF, and 1 mM Na_3_VO_4_. Cell lysates were incubated on ice and vortexed for 15 s in four intervals of five minutes. Samples were centrifuged at 13,000 rpm at 4 °C for 5 min. The supernatants and total EV fractions were collected for protein quantification and downstream analysis. Samples were submitted to SDS-PAGE electrophoresis in Any kD™ Criterion™ TGX Stain-Free™ Protein Gels (Ref. 5678124; Bio-Rad, Hercules, FL, USA) in reducing or non-reducing conditions and transferred to polyvinylidene difluoride membranes. The membranes were incubated with primary antibodies and the corresponding secondary antibodies were coupled to horseradish peroxidase. Protein content was revealed with a Clarity™ Western ECL substrate (Ref. 170-5061; Bio-Rad, Hercules, FL, USA). Images were acquired in a ChemiDoc™ Touch Imaging System and analyzed with Image Lab software. The primary antibodies were as follows: CD63 antibody (mouse IgG1κ) (Ref. NBP2-42225, Novus Biologicals, Centennial, CO, USA), Human Annexin V antibody (mouse IgG2A) (Ref. MAB3991, R&D Systems, Minneapolis, MN, USA), GM130/GOLGA2 antibody (rabbit IgG) (Ref. NBP2-53420, Novus Biologicals), Human GRP78/HSPA5 antibody (mouse IgG2b) (Ref. MAB4846, R&D Systems), and TSG101 antibody (mouse IgG1) (Ref. NB200-112, Novus Biologicals). Secondary antibodies included: Goat anti-rabbit IgG-HRP (Ref. sc-2357, Santa Cruz Biotechnology, Santa Cruz, CA, USA), m-IgGκ BP-HRP (Ref. sc-516102, Santa Cruz Biotechnology), Goat anti-mouse IgG1-HRP Human (Ref. 1070-05, Southern Biotech, Birmingham, AL, USA), Goat anti-mouse IgG2a-HRP Human (Ref. 1080-05, Southern Biotech), and Goat anti-mouse IgG2b-HRP Human (Ref. 1090-05, Southern Biotech).

### 2.14. Cryo-Electron Microscopy (Cryo-EM)

The EV size and shape analysis was conducted by cryo-EM. Samples (4 µL) were adhered to QUANTIFOIL^®^ holey carbon (2/1) copper grids (Ref. Q225CR-06; Quantifoil Micro Tools GmbH, Großlöbichau, Germany) after glow discharge with argon plasma. Next, grids were blotted to remove the liquid excess and vitrified in ethane in a Leica Automatic Plunge Freezer EM GP2 (Leica, Wetzlar, Germany). Samples were maintained in liquid nitrogen and detected using a JEM-2200FS/CR transmission electron microscope (JEOL Ltd., Akishima, Japan).

### 2.15. mRNA Sequencing

Libraries were prepared from 350 ng of total RNA using the QuantSeq 3′ mRNA-Seq Library Prep Kit FWD for Illumina with unique dual indices (Ref. 114, Lexogen GmbH, Vienna, Austria). The RNA concentration was measured with the Qubit^®^ RNA Assay (Ref. Q32852, Thermo Fisher Scientific) and the quality was assessed using the RNA ScreenTape Assay (Ref. 5067-5576, Agilent Technologies, Santa Clara, CA, USA) for the TapeStation System 4150 (Ref. G2992AA, Agilent Technologies). All samples showed a RIN above 9.8. First, the cDNA strand was synthesized using oligodT from total RNA. Then, the RNA was degraded, and second strand synthesis was performed by random priming. After bead purification, adaptor ligation and unique dual index incorporation (Set UDI12B_0001-0096, Lexogen GmbH) was performed by endpoint PCR, according to the following protocol: 30 s at 98 °C, 11–25 cycles at 98 °C for 30 s, 65 °C for 20 s, and 72 °C for 30 s, and a final extension at 72 °C for 1 min. The optimal amplification cycle was empirically determined for each sample using the PCR-Add-on-Kit (Ref. 020.96, Lexogen GmbH). Once amplified, libraries were bead-purified and subjected to quality control. Quantification was performed with the High Sensitivity DNA Qubit^®^ Assay. Next, the size distribution of the libraries was assessed with the High Sensitivity D100 ScreenTape Assay (Ref. 5067-5584, Agilent Technologies) for TapeStation System 4150. The average library size was 318 bp and no overamplification or adaptor dimers were detected. 

### 2.16. mRNA Sequencing Data Analysis

Primary analysis was conducted with Illumina DRAGEN FASTQ Generation online software (v. 3.8.4). Secondary analysis, concerning alignment, mapping, and differential expression analysis, was carried out using the Bluebee^®^ Quantseq pipeline. Briefly, reads were quality and adapter trimmed and mapped to the human genome version GRCh38 using STAR Aligner with modified ENCODE settings [23]. HTSeq-count was used for gene read counting and differential expression analysis (DEA) was performed using the DESeq2 method [24] (GEO accession number GSE201695). Principal component analysis was conducted with R Studio using the packages BiocManager, DESeq2, and ggplot2 (Appendix A). Heatmaps were constructed using R Studio with ComplexHeatmap, viridis, and dplyr packages. GO terms and KEGG pathways enrichment analysis were performed with a DAVID online tool [22] using all genes with counts > 0 as the background. Additionally, to interpret the genomic data, gene set enrichment analysis (GSEA) was performed [25,26]. Biological connections among selected targets were estimated using Search Tool for the Retrieval of Interacting Genes/Proteins (STRING) [27]. STRING analysis was performed with medium confidence (interaction score 0.4) only for the query proteins. 

Sequencing data were provided as fold-change (FC) values with base mean counts, *p*-value, and adjusted *p*-value (*p*-adj). FC values were calculated as: (counts in the experimental condition)/(counts in the control condition). In order to select direct and indirect targets of each miRNA studied, Venn diagrams were generated using cut-offs with a base mean > 30 and log_2_FC > 0.5 (up-regulation) or <−0.5 (down-regulation).

### 2.17. Statistical Analysis

All quantitative variables were expressed as the mean ± SEM for the laboratory or in vitro data and with a median and interquartile range [IQR 25th–p75th] for the clinical data; categorical or ordinal variables are expressed by the absolute frequencies and percentages in all cases.

In vitro data were compared using the analysis of variance (ANOVA) with the least significant difference’s test as a post hoc multiple comparison analysis (homogeneity of variances). 

The normal distribution of the variables was assessed with the Kolmogorov–Smirnov (n > 50) test or the Shapiro–Wilk (n < 50) test. For time-to-event variables, the median and the 95% CI was analyzed with the Kaplan–Meier method. To study the relationship of the clinical variables, we used the log-rank. Eventually, hazard ratios and their 95% CI were estimated using univariate and multivariate Cox regression models, adjusting according to the relevant clinical factors (none, Child–Pugh classification, and performance status) and stratified for the baseline BCLC staging. Clinical data on the BCLC cohort were analyzed using SAS software version 9.4 (SAS Institute, Cary, NC) on Hospital Clínic de Barcelona, the correlation of miRNA expression with clinical data was conducted with SPSS statistical software v. 11.0 (SPSS Inc., Chicago, IL, USA) and R Studio using the survival package by Hospital Virgen del Rocío. All tests were two-sided with a significant level of 0.05 (for all plots, the level of significance was expressed as: * *p* ≤ 0.05, ** *p* ≤ 0.01, *** *p* ≤ 0.001, and **** *p* ≤ 0.0001). Graphs were performed using the GraphPad Prism software v6.0 (GraphPad Software Inc., San Diego, CA, USA).

## 3. Results

### 3.1. Analysis of miRNA Expression in Primary Human Hepatocytes and Liver Cancer Cells

To screen for candidate biomarkers, we profiled miRNA expression in the Sorafenib-treated HepG2 cells and primary human hepatocytes. The hierarchical clustering of miRNA expression confirmed the separation between HepG2 and primary human hepatocytes. The control and Sorafenib-treated HepG2 cells also clustered separately, indicating that Sorafenib exerts its anti-tumoral properties in a time-dependent and cell-specific manner (Figure 1A and Appendix A). We found 9 down-regulated and 24 up-regulated miRNAs in the HepG2 cells compared with the primary hepatocytes, whose targets were associated with cancer related processes (Figure 1B,D and Appendix A). In the primary hepatocytes, Sorafenib (24 h) down-regulated miR-376a-3p and up-regulated miR-29c-3p and miR-671-3p (Figure 1C,D and Appendix A).

The administration of Sorafenib involved sustained ER stress associated with a transition from an autophagic survival phase (6–12 h) to apoptosis (24 h) in HepG2 cells [28]. In tumor cells, Sorafenib down-regulated the expression of miR-551a (fold-change = 0.377; *p* = 0.010), and up-regulated miR-122-5p (fold-change = 3.257; *p* = 0.037), miR-200c-3p (fold-change = 2.145; *p* = 0.031), and miR-505-5p (fold-change = 2.528; *p* = 0.031) (6 h) (Figure 2A). After 24 h of treatment, Sorafenib altered the expression of seven miRNAs (Figure 2B). This epigenetic signature was defined as the down-regulation of miR-148b-3p (fold-change = 0.442; *p* = 0.011), miR-194-5p (fold-change = 0.349; *p* = 0.042), miR-222-5p (fold-change = 0.189; *p* = 0.020), and miR-512-3p (fold-change = 0.465; *p* = 0.035), and up-regulation of miR-27a-3p (fold-change = 4.118; *p* = 0.033), miR-193b-3p (fold-change = 4.403; *p* = 0.020) and miR-375 (fold-change = 2.457; *p* = 0.025) (Figure 2B). A total of 2552 targets were predicted (Appendix A). The enriched terms were coherent with the cancer related pathways and kinase inhibitor activity (Figure 2C,D and Appendix A). 

### 3.2. miRNA Expression Profile in Sorafenib-Treated Mouse HCC Xenografts

The alteration in the miRNA expression pattern induced by Sorafenib was also evaluated in a mouse xenograft model based on the subcutaneous injection of HepG2. Sorafenib reduced tumor volume in nude mice (Figure 2E). The miRNA expression pattern obtained in HepG2 cells treated with Sorafenib was tested in the tumor explants after Sorafenib treatment and compared to the control mice. In concordance with the in vitro study, miR-27a-3p, miR-193b-3p, miR-200c-3p, and miR-505-5p were up-regulated, whereas miR-194-5p was down-regulated in tumors from the Sorafenib-treated mice compared to the control mice (Figure 2F). Although they were not causative of tumor reduction, they were clearly related to the reduction in the tumor volume induced by Sorafenib. Thus, they constitute a miRNA signature of treatment response in vivo.

### 3.3. Functional Analysis of Differentially Expressed miRNAs in Sorafenib-Treated HepG2 Cells 

The corresponding miRNA mimics and antisense oligonucleotides were used for the functional analysis of down-regulated or up-regulated miRNAs, respectively. The validation of the down- or up-regulated, and miRNA transfections are shown in Appendix A. 

Table 1 summarizes the effect of miRNAs seen in the functional studies (Figure 3). miR-148b-3p and miR-512-3p were shown to increase proliferation in HepG2 cells (Figure 3A). Therefore, the anti-proliferative properties of Sorafenib were related to the down-regulation of miR-148b-3p and miR-512-3p. Caspase-3 activity was reduced by miR-122-5p and miR-505-5p, while it was increased by miR-222-5p (Figure 3C,D). Regarding the metastatic potential of miRNAs in liver cancer cells, miR-194-5p, miR-512-3p, and miR-551a promoted (Figure 3E), whereas miR-27a-3p, miR-193b-3p, and miR-200c-3p decreased cell migration (Figure 3F). Thus, the effect of Sorafenib on miRNA expression exerted an anti-tumoral role by decreasing migration in the HepG2 cells. miR-27a-3p, miR-200c-3p, and miR-551a reduced, whereas miR-148b-3p, miR-193b-3p, miR-194-5p, and miR-222-5p increased the cell invasiveness (Figure 3G,H). Hence, the up-regulation of miR-27a-3p and miR-200c-3p, and down-regulation of miR-148b-3p, miR-194-5p, and miR-222-5p in the Sorafenib-treated liver cancer cells had a beneficial, anti-tumoral effect. In conclusion, miR-27a-3p and miR-200c-3p were considered anti-tumoral miRNAs, whereas miR-122-5p, miR-148b-3p, miR-194-5p, miR-222-5p, miR-505-5p, and miR-512-3p exerted pro-tumoral activities (Table 1).

### 3.4. Sorafenib Reduced EVs Secretion in HepG2 Cells

Liver cancer cells have been shown to alter the secretion of EVs that might modify treatment response and promote resistance [29,30]. At 6 h of Sorafenib treatment, we did not find differences in the size or concentration of EVs in the culture medium. However, Sorafenib increased their size at 24 h, which was more prominent in the fraction of large EVs (Figure 4A). The measurement of the expression of CD63 and Tgs101 was a useful marker of exosomes in EVs. Sorafenib (24 h) decreased the expression of CD63 and Tgs101, suggesting that sorafenib decreased the release of exosomes in the culture medium (Figure 4B). No expression of ER (Grp78), plasma membrane (Annexin V), Golgi apparatus (GM130), or mitochondria (PHB, prohibitin) markers were observed in the exosomes (Figure 4B). The presence of Annexin V in large EVs suggests that Sorafenib might induce the release of apoptotic bodies (Figure 4B). cryo-EM analysis confirmed that Sorafenib shifted the population toward larger EVs (Figure 4C,D). Furthermore, Sorafenib diminished EV secretion and greatly increased the accumulation of protein aggregates (Figure 4C,E)

Next, we assessed the pattern of miRNA secretion within EVs (Figure 4F,G). At 6 h of treatment, miR-122-5p and miR-200c-3p accumulated in very small EVs (Figure 4F). The comparison of the miRNA levels in the released EVs and cells showed that miR-505-5p was highly regulated by Sorafenib (Appendix A). Similarly, at 24 h, Sorafenib increased miR-27a-3p, miR-148b-3p, miR-193b-3p, miR-194-5p, and miR-375 expression in the very small EVs fraction (Figure 4G). The comparison between miRNA levels in the EVs and cells pointed out that Sorafenib greatly induced the secretion of miRNAs at 24 h (Appendix A). 

### 3.5. Expression of Circulating miRNAs as Biomarkers of Disease Prognosis

The usefulness of the miRNA profile as a circulating biomarker of HCC prognosis and Sorafenib treatment response was tested in two independent cohorts. miRNA expression was assessed before treatment initiation and in a time-dependent manner (baseline and 1-month levels). We examined miRNAs in a study cohort from Hospital University “Virgen del Rocío” (n = 36) and then assessed this signature in a larger validation cohort from the Barcelona Clinic Liver Cancer (n = 81). Given the great impact of BCLC stage on patient prognosis, we segmented patients according to the cancer stage (BCLC-B or BCLC-C stage) to perform further analysis. Time-to-event regression models were constructed using critical clinical endpoints such as death and quick progression rates at 30 and 60 days (Appendix A). 

Although different miRNAs have been associated with survival or progression, only results regarding the baseline miR-200c-3p and time-dependent levels of miR-222-5p and miR-512-3p were related to survival in both cohorts in the BCLC-C stages (Figure 5, Appendix A). In the study cohort, the baseline levels of miR-200c-3p correlated with a lower risk of death (HR = 0.7710; 95% CI 0.5994–0.9917), confirming previous results about its anti-tumoral properties in in vitro functional studies (Table 1). On the other hand, time-dependent values of miR-222-5p (HR 1.0830; 95% CI 1.0078–1.1640) and miR-512-3p (HR 1.0660; 95% CI 1.0095–1.1260) were associated with the increased probability of patient death (Figure 5A and Appendix A). The validation cohort provided support to these results. The pro-tumoral roles of time-dependent levels of miR-222-5p (HR 40.1822; 95% CI 5.5213–292.4300) and miR-512-3p (HR 50.1449; 95% CI 1.5718–1599.7400) were validated in this cohort, whilst the baseline levels of miR-200c-3p were related to better prognosis (HR 0.2205; 95% CI 0.05046–0.9632) (Figure 5B, and Appendix A). These clinical data are strongly aligned to the anti-tumoral properties of miR-200c-3p and the pro-tumoral role of miR-222-5p and miR-512-3p in the functional studies described above. However, due to the wide confidence intervals for miR-512-3p in both cohorts, the results regarding this miRNA should be taken with caution.

### 3.6. Alteration in the mRNA Expression Pattern Induced by Selected miRNAs

To look for direct or indirect targets of miR-200c-3p, we searched in those genes that were up-regulated after miRNA inhibition (Figure 6A,B and Appendix A). The enrichment analysis showed alterations in transcriptional regulation, membrane transporters (*SLC7A11*, *ATP1B1*), and metallothionein (MT) expression (*MT1B*, *MT1M*, *MT1A*) (Figure 6C–E and Appendix A). The GSEA analysis showed that miRNA inhibition increased the expression of genes related to the negative regulation of cell death (*PIM3*) compared to the transfection control, meaning that miR-200c-3p could be functioning as a pro-apoptotic factor (Figure 6D). STRING analysis predicted 191 edges (protein–protein interaction or PPI enrichment 1.66 × 10^−6^) and allowed for the clustering of target genes (Figure 6E and Appendix A). It is important to mention that cluster 2 included processes such as response to oxidative damage or misfolded protein binding, involving genes such as *HSPA1B* or *NFE2L2* (Figure 6F and Appendix A). 

miR-222-5p was herein considered to be a pro-tumoral miRNA. We identified mRNA expression down-regulated by miR-222-5p mimics (M-miR-222-5p) but not regulated in the transfection control in the mRNA-Seq experiments (Figure 7A,B and Appendix A). Curiously, mitochondrially encoded tRNA-Phe (*MT-TF*) and cytochrome c oxidase III (*MT-CO3*) are among the most significantly regulated terms. miR-222-5p altered cell cycle progression, disturbed cell proliferation and death, and cell metabolism (Figure 7C). GSEA confirmed the regulation of these processes (Figure 7D). The STRING analysis showed 4286 nodes with PPI enrichment of 1 × 10^−16^. Gene expression was grouped into six clusters (Figure 7E, Appendix A). Importantly, cluster 2 included a wide variety of genes associated with growth factor responses as well as reduced apoptosis induction mechanisms (*FAS*, *SOX9*, *TGFB1*, *AXL*, *IRS1*, *SRC*, etc.). Lipid (cluster 3) (*ABCG8*, *SERPINC1*, *ACSL4*, *HMGCR*, *PLA2G15*, *FADS3*, *FADS2*, *ELOVL6*, etc.) and carbon metabolism (cluster 4) (*FH*, *ALDOC*, *ENO3*, *UGP2*, etc.) were reduced after the miRNA mimic. Cholesterol homeostasis was negatively impacted by miR-222-5p as well as fatty acid biosynthesis. Regarding cell cycle (cluster 5), reduced expression of *RB1* or *CCNB1* were observed (Figure 7F and Appendix A).

As in the case of miR-222-5p, to look for targets regulated by miR-512-3p, we selected genes that were down-regulated with the miRNA mimic but not regulated in the transfection control (Figure 8A,B and Appendix A). These targets were related to cofactor synthesis or oxidant response. The miR-512-3p effects seemed to be focused on mitochondria (Figure 8C). GSEA was also related to miR-512-3p with other downregulated processes such as telomere organization (Figure 8D). STRING analysis showed 8164 nodes with PPI enrichment 1 × 10^−16^. Gene expression was grouped into six clusters (Figure 8E and Appendix A). Of particular interest, in cluster 4, we found extensive signaling induced by the regulation of small GTPases (*SOS2*, *ARHGEFs*, *RAB3D*, *RAP2C*, etc.). This could be connected with endosomal transport in the GSEA analysis and autophagy initiation (*ATG16L1*, *ATG14*, *WIPI1*). Cluster 5 contained cellular oxidant detoxification systems (*MAOB*, *ADH4*, *ADH5*, *GPX3*, *GSR*, *APOM*, *HP*, *MSTG1-3*, etc.). Metabolic alterations mediated by miR-512-3p were more associated with oxidative metabolism and mitochondrial organization (cluster 6) (*MRPLs*, *MRPSs*, *NDUFAF1*, etc.) (Figure 8F and Appendix A). 

Finally, common regulation induced by the down-regulation of miR-200c-3p, and up-regulation of miR-222-5p and miR-512-3p involves the up-regulation of *SNAI1* or *CADH6*.

## 4. Discussion

The study identified the miRNA signature related to sorafenib responsiveness in patients with advanced HCC useful for personalized medicine and clinical decision making. Over 2000 different mature miRNAs have been identified thus far (The miRBase Sequence Database-Release 19.0), which accounts for approximately 5% of the transcribing genome [31]. The miRNA profiling of primary human hepatocytes and HepG2 cells revealed a total of 33 differentially expressed miRNAs (Figure 1A,B and Appendix A). Bioinformatic analysis confirmed their relationship with tumor initiation and progression including KEGG pathways such as colorectal cancer or the regulation of stem cell properties. The usefulness of tumor non-coding RNAs as biomarkers of liver cancer that can act as oncogenes or tumor suppressors has previously been proposed. A study comparing neoplastic HCC nodules and cirrhotic tissues displayed 62 altered miRNAs [32]. Six of these tumor-altered miRNAs (miR-193a-5p, miR-339-5p, miR-375, miR-483-5p, miR-532-3p, and miR-660-5p) were also differentially expressed in our study related to Sorafenib effectiveness (Figure 1B and Appendix A). The anti-tumoral role of miR-375 has been validated by other authors. In fact, the administration of miR-375 mimics has been able to suppress the growth of hepatoma xenografts in nude mice [33]. Moreover, the loss of the liver specific miR-122-5p correlates with HCC progression [34]. miRNA profiling in primary human hepatocytes have identified a signature based on the expression of miR-122-3p/-5p, miR-192-3p/5, miR-148b-3p, miR-193b-3p, miR-375, miR-215-5p, miR-194-5p, miR-885-5p, miR-23b-3p, and miR-4800-5p as a specific profile of primary human hepatocytes [35]. The enrichment of miR-122 and miR-375 confirmed their anti-tumor properties, as previously described in the literature, and both were up-regulated in response to Sorafenib in our experimental setting. Furthermore, our results coincide with some of the miRNAs highlighted in this last study. In the relationship with lipid and drug metabolism, miR-149-5p was up-regulated in response to chenodeoxycholic acid [36], showing a similar trend in our study, indicating that pro-lipogenic pathways could be associated with hepatocarcinogenesis. Precisely, a shift in the lipid profile of primary human hepatocytes has been related to miR-27a and miR-21 and down-regulated miR-30 during their dedifferentiation in cell culture [37]. 

We obtained a ncRNA profile of Sorafenib response based on the expression of 11 miRNAs. To assess how they might influence disease outcome, we investigated their transfer in EVs in vitro and as circulating biomarkers in plasma. The sensitivity of HepG2 to Sorafenib is associated with the transference of miRNA signatures, among other components, in exosomes. Liver cancer cell-derived exosomes are able to induce Sorafenib resistance in vivo and in vitro by suppressing apoptosis through the regulation of the HGF/c-Met/Akt pathway [29]. Few studies have obtained reliable data on the use of miRNA circulating biomarkers in patients treated with Sorafenib. Although low levels of miR-122-5p and high levels of miR-miRNA-21 and miRNA-96 in EVs and plasma have been tested as diagnostic or potential prognostic biomarkers in HCC, no risk models were performed in this study as well as being not related to the effectiveness of therapy [38]. Another study evaluated the longitudinal expression of miR-23b-3p and miR-126-3p in a cohort of seven patients under Sorafenib treatment without establishing any correlation with survival or progression [39]. miR-31-5p, miR-221, miR-30e-3p, or miR-30d have been related to Sorafenib resistance in cultured renal and liver cancer cells [40,41,42,43]. However, no robust in vitro, in vivo, and clinical studies have been developed to identify a miRNA signature useful to predict disease outcome and treatment response in advanced HCC. Here, we provide Cox regression models that accurately validate the prognostic role of miR-200c-3p, miR-222-5p, and miR-512-3p in two different populations (Figure 5A,B).

Of note, the upregulation of miR-200c-3p (6 h, Figure 2A) by Sorafenib has been shown to be coherently related to the anti-tumoral properties in HepG2 and its plasma levels were associated with increased survival (Figure 5A,B). The anti-tumoral role of miR-200c-3p has been related to reduced metastasis in the colorectal xenograft model [44] and reduces the infiltration capacity of macrophages cocultured with breast cancer cells [45]. Here, we additionally demonstrated that miR-200c-3p regulated the damage response by altering *HSPA1B* or *NFE2L2* (Figure 6). HSPA1B is a chaperone that regulates protein folding. Higher accumulation of unfolded proteins could lead to higher ER stress, an early event during Sorafenib response. Therefore, we correlate here the augmentation of unfolded protein response mechanisms with the response to Sorafenib. In fact, miR-200c-3p has been shown to regulate ER stress in prostate cancer [46]. Additionally, we have recently shown that the downregulation of Nrf2 (*NFE2L2*) and thioredoxin-1 are associated with the anti-tumoral activity of Sorafenib in cultured HepG2, and in a xenograft model [47]. Nrf2 has also been shown to have an important role in ferroptosis inhibition upon sorafenib treatment as a mechanism of resistance through the expression of MTs [48]. We identified several MTs including MT1B or MT1M as highly expressed upon miR-200c-3p inhibition. Hence, it seems that heavy metal detoxification and antioxidants might be regulating resistance in our setting. Importantly, PIM3 is a proto-oncogene with serine/threonine kinase activity that is usually up-regulated in HCC. Furthermore, in mice, PIM3 is able to accelerate hepatocarcinogenesis induced by diethylnitrosamine [49]. Not only that, but PIM3 controls intracellular cascades regulating migration and invasion [50] as well Bcl-2 related anti-apoptotic proteins [51]. All in all, in light of these previous studies, we can conclude that signaling induced by miR-200c-3p inhibition is potentially pro-tumoral, anti-apoptotic, and pro-resistance, supporting the good prognosis value of miR-200c-3p in the plasma of patients treated with Sorafenib.

On the other hand, the overexpression of miR-222 has been found in liver cancer [52]. The administration of the miR-222 mimic plays a crucial role in promoting cell proliferation, migration, and invasion, and decreases cell apoptosis in HCC cells [53]. In this sense, the down-regulation of miR-222-5p exerted anti-apoptotic properties in Sorafenib-treated HepG2 cells (Table 1). According to this result, miR-222-5p has been found as one of the key biomarkers related to death in patients (Figure 5A,B). Sequencing studies revealed that miR-222-5p suppressed cell cycle and reduced cell lipid and carbon metabolism (Figure 7). However, mitochondrial functionality seemed to be significantly enriched due to the down-regulation of the *MT-TF* and *MT-CO3* transcripts. Deregulation of mitochondria and impaired respiration have been linked with HCC, in particular, the loss of mtDNA [54,55]. Therefore, the down-regulation of mitochondrial components might constitute a pro-survival mechanism. Moreover, we observed a down-regulation of tyrosine kinases and growth receptor related pathways that include AXL, IRS1, or SRC. The reduction in Sorafenib targets has been described as a potential resistance mechanism, in particular, the reduction in phosphorylation of the Ras pathway. Therefore, our results support this idea as a mechanism of plausible resistance and reduced response to Sorafenib [56]. Pro-tumoral effects of miR-222-5p were supported by the downregulation of several key enzymes of metabolism. For instance, low expression of UGP2 and ENO3 have been related to poor outcome in HCC [57,58]. In addition, UGP2 has been linked to fatty acid metabolism, supporting our results that indicate that lowered lipid metabolism is related to miR-222-5p poor outcome [57]. Regarding cell cycle, one of the key regulators of cell cycle progression is RB1, which has been characterized as a target of miR-222-5p. By means of regulating RB1, miR-222-5p contributes to migration and invasion, and also to altered lipid metabolism [59]. Additionally, miR-222-5p repressed SOX9, whose down-regulation during YAP signaling has been proposed as a mechanism of malignancy in HCC [60]. All in all, these findings underwrite the pro-tumoral properties of miR-222-5p during Sorafenib therapy response. 

Sorafenib also induced miR-512-3p down-regulation in our study (24 h, Figure 2B). The expression of miR-512-3p has been shown to be increased in HCC tissues compared with adjacent non-tumor tissues obtained in patients submitted to liver resection for VHB-related HCC [61]. It has also been associated with a pro-tumoral activity characterized by cisplatin-resistant phenotype in human germ cell tumors [62]. Our functional studies showed that the administration of the miR-512-3p mimic increased cell proliferation and cell migration (Table 1). In fact, increasing expression of this miRNA could be correlated with poor prognosis in BCLC-C stage patients (Figure 5A,B). Oxidative metabolism was one of the items controlled by miR-512-3p (Figure 8). In particular, we observed the down-regulation of several mito ribosomal proteins such as MRPS22 or MRPS36, among others. MRPS31 loss has been identified as a poor outcome biomarker in HCC, behaving as a driver of malignancy and showing prognostic value [63]. This miRNA blocked mitochondrial metabolism, mitochondrial complex assembly, and, possibly as a consequence, oxidant response. For instance, glutathione peroxidase 3 (GPx3) has been correlated with a higher degree of invasion and tumor stage after liver resection in HCC [64]. Additionally, although in a different context, miR-512-3p has been found to regulate oxidative damage in endothelial dysfunction [65]. Hence, mitochondrial organization and oxidative metabolism are key processes related to miR-512-3p upregulation and poor outcome after Sorafenib treatment. Another crucial factor in cancer is autophagy induction. This double-edged process might be beneficial or detrimental for patient prognosis, and is highly dependent on the context of liver cells [66]. We have previously shown that autophagy induction is necessary to initiate cell death induced by Sorafenib [28]. Therefore, downregulation of ATG16L1, ATG14, or WIPI could constitute a pro-tumoral signature in this specific context. Eventually, as a common mechanism induced by miR-200c-3p, miR-222-5p, and miR-512-3p, the *SNAI1* and *CADH6* transcripts were upregulated in response to miR-200c-3p inhibition, and miR-222-5p and miR-512-3p upregulation. Both of them have been related to epithelial-to-mesenchymal transition in HCC [67,68].

## 5. Conclusions

In summary, our studies revealed that the HepG2 cells displayed a specific miRNA profile different to that of the primary human hepatocytes upon Sorafenib treatment. Sorafenib induced a miRNA signature composed of 11 miRNAs (miR-27a-3p, miR-122-5p, miR-148b-3p, miR-193b-3p, miR-194-5p, miR-200c-3p, miR-222-5p, miR-375, miR-505-5p, miR-512-3p, and miR-55a). Functional studies revealed that miR-27a-3p and miR-200c-3p exerted anti-tumoral activities, whereas miR-122-5p, miR-148b-3p, miR-194-5p, miR-222-5p, miR-505-5p, and miR-512-3p promoted pro-tumoral actions. We also assessed the trafficking of miRNAs in the released EVs by HepG2, showing the accumulation of miR-27a-3p, miR-122-5p, miR-148b-3p, miR-193b-3p, miR-194-5p, miR-200c-3p, and miR-375 in the Very Small EVs fraction. In the nude mice experiments, miR-27a-3p, miR-193b-3p, miR-194-5p, miR-200c-3p, and miR-505-5p were related to tumor reduction after Sorafenib treatment. Among the identified miRNAs in the functional studies, the miRNA signature constituted of miR-200c-3p, miR-222-5p, and miR-512-3p showed a high relevance in the clinical setting. Whilst miR-200c-3p was related to increased survival, miR-222-5p and miR-512-3p were associated with poor prognosis. 

## Figures and Tables

**Figure 1 cells-11-02673-f001:**
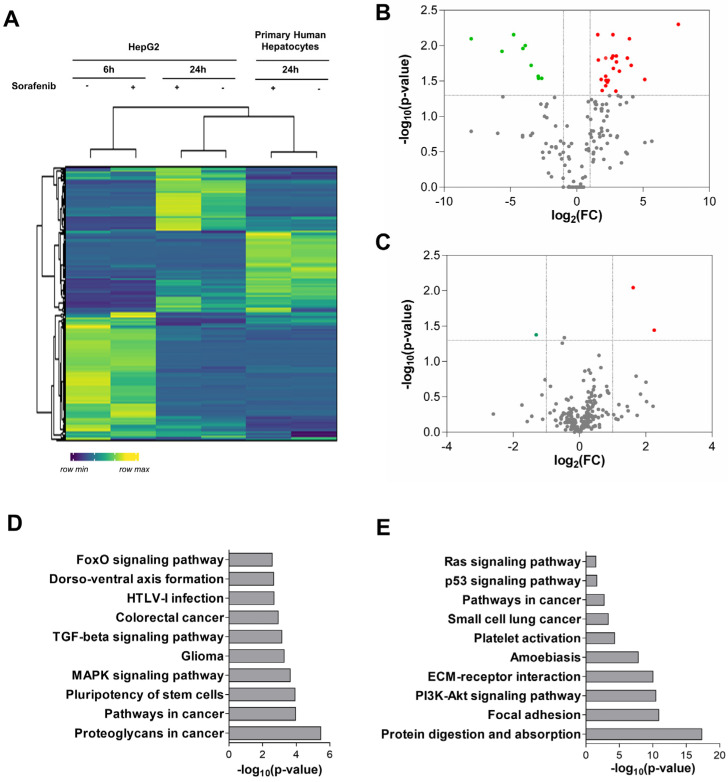
miRNA pattern induced in primary human hepatocytes. (**A**) Heatmap of miRNA expression in HepG2 cell lysates at 6 and 24 h of treatment, compared to primary hepatocyte expression profile after Sorafenib (24 h) (n = 3). (**B**) Volcano plot of miRNA expression analysis in the HepG2 cells compared to primary human hepatocytes (non-treated) (n = 3). Up-regulated miRNAs are shown in red and down-regulated miRNAs are shown in green. (**C**) Volcano plot of differentially expressed miRNAs in primary human hepatocytes treated with Sorafenib (24 h) (n = 3). (**D**) Top 10 most significant KEGG pathways enriched in targets of miRNAs differentially expressed in HepG2 cells. (**E**) Top 10 most significant KEGG pathways enriched in targets of miRNAs differentially expressed in treated primary human hepatocytes.

**Figure 2 cells-11-02673-f002:**
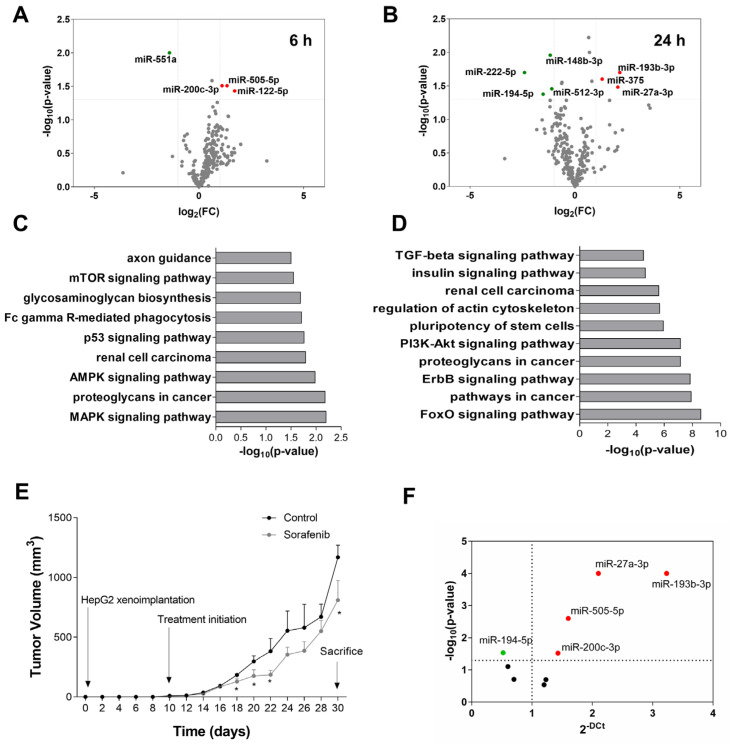
miRNA expression profile induced by Sorafenib in cellular and mouse models. (**A**) Volcano plot of miRNA expression analysis in cell lysates at 6 h of treatment (n = 3). Up-regulated miRNAs are shown in red and down-regulated miRNAs are shown in green. (**B**) Volcano plot of miRNA expression analysis in cell lysates at 24 h of treatment (n = 3). Up-regulated miRNAs are shown in red and down-regulated miRNAs are shown in green. (**C**) Top 10 most significant KEGG pathways enriched in the targets of miRNAs differentially expressed at 6 h of treatment. (**D**) Top 10 most significant KEGG pathways enriched in the targets of miRNAs differentially expressed at 24 h of treatment. (**E**) Tumor volume monitoring during Sorafenib and vehicle treatments. (**F**) Differentially expressed miRNAs in the tumors of subcutaneous mouse models treated with Sorafenib (n = 14).

**Figure 3 cells-11-02673-f003:**
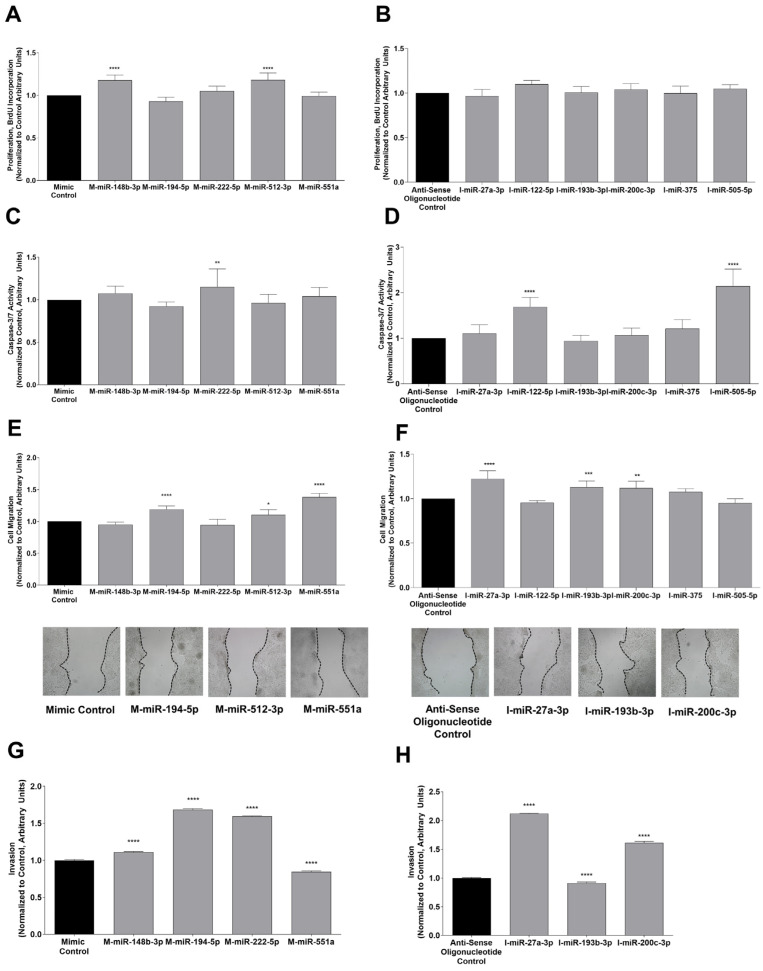
Functional analysis with mimics (left panel) and inhibitors (right panel) of differentially expressed miRNAs in HepG2 cells treated with Sorafenib. Proliferation (mimics **A**, inhibitors **B**), caspase-3/7 activity (mimics **C**, inhibitors **D**), migration (mimics **E**, inhibitors **F**), and invasion (mimics **G**, inhibitors **H**) were tested to unravel anti- or pro-tumoral effects of miRNAs after Sorafenib treatment. Invasion was tested for the miRNA mimics and inhibitors that showed the upregulation of cell migration with *p*-value < 0.01, or those that downregulated cell migration more that 5%. * *p* ≤ 0.05, ** *p* ≤ 0.01, *** *p* ≤ 0.001, and **** *p* ≤ 0.0001 between the transfection control and mimic or inhibitor tested.

**Figure 4 cells-11-02673-f004:**
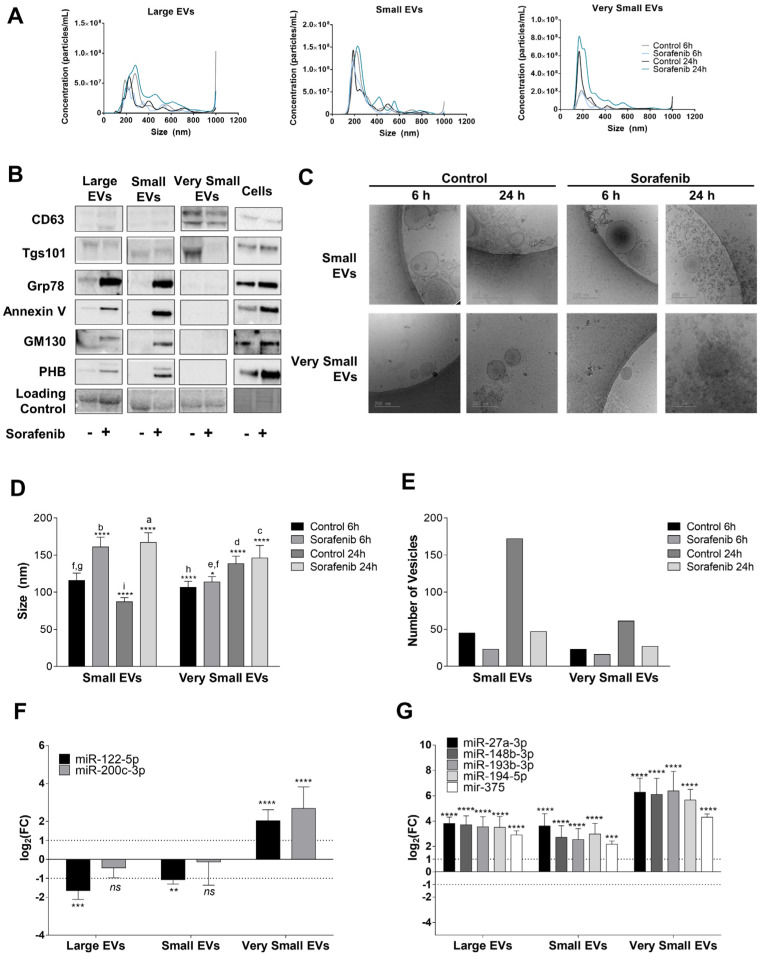
Extracellular vesicle secretion induced by Sorafenib and their miRNA content. (**A**) Particle size and concentration analysis of Large, Small, and Very Small EVs obtained at 6 and 24 h. (**B**) Expression of EV markers and cellular contaminants in Large, Small, and Very Small EVs and cell lysates obtained at 24 h after Sorafenib treatment. Total lane protein content was used as the loading control. (**C**) Representative images of cryo-EM of Small and Very Small EVs obtained at 6 and 24 h from the control and Sorafenib-treated cells. (**D**) Assessment of EV size (nm) in the cryo-EM images. (**E**) Number of EVs quantified in the cryo-EM images. (**F**) miRNA expression in the three fractions of EVs at 6 h. (**G**) miRNA expression in the three fractions of EVs at 24 h. Fold-change values were calculated between Sorafenib and the control treated samples. Results are expressed as the mean ± SEM of six independent experiments. Ns, non-significant; * *p* ≤0.05, ** *p* ≤ 0.01, *** *p* ≤ 0.001, and **** *p* ≤ 0.0001 between the miRNA expression in the control and Sorafenib derived EVs. Multiple comparison test statistics are expressed with lower case a–i.

**Figure 5 cells-11-02673-f005:**
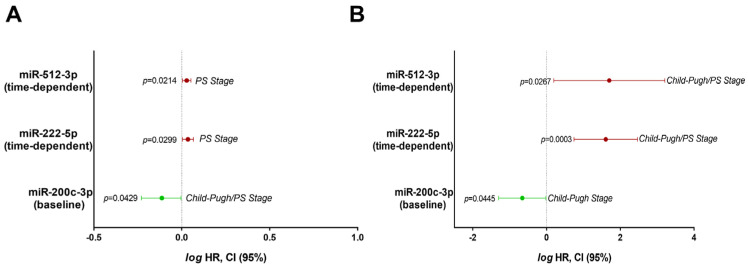
Hazard ratio models of circulating miRNAs coincident in two independent patient cohorts as prognostic factors of patient death. (**A**) Death risk models of miR-200c-3p (baseline levels), miR-222-5p (time-dependent values), and miR-512-3p (time-dependent values) in the study cohort (BCLC-C, n = 27). (**B**) Death risk models of miR-200c-3p (baseline levels), miR-222-5p (time-dependent values), and miR-512-3p (time-dependent values) in the validation cohort (BCLC-C, n = 47). Results are expressed in the logarithmic scale as the hazard ratio and 95% confidence intervals.

**Figure 6 cells-11-02673-f006:**
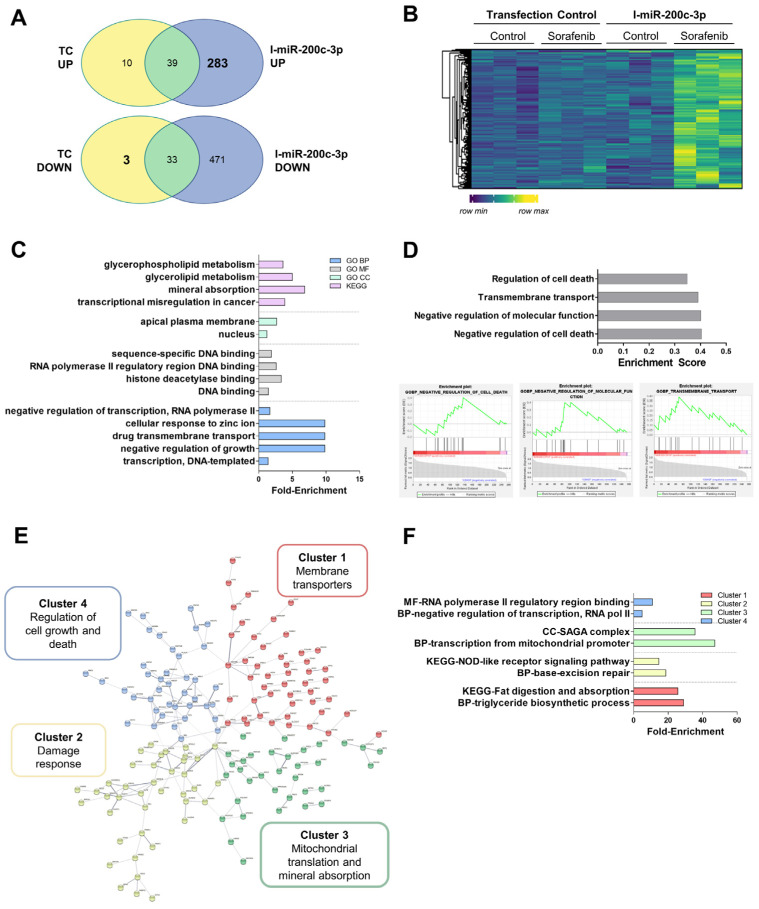
RNA-sequencing data of miR-200c-3p down-regulation in HepG2 cells showed increased damage response mechanisms. (**A**) Venn diagrams showing the mRNAs regulated upon miR-200c-3p inhibition (I-miR-200c-3p) compared to the transfection control (TC). (**B**) Heatmap of genes regulated by miR-200c-3p. (**C**) Ffold-enrichment values of the top five most significantly enriched GO BP, MF, CC, and KEGG pathways. (**D**) Top 10 significant hallmarks GSEA up-regulated after I-miR-200c-3p. Plots were provided for the three most enriched terms. (**E**) Diagram showing the clustering of STRING analysis. (**F**) Fold-enrichment values of the top two most significantly enriched terms in either the GO BP, MF, CC, or KEGG pathways in each STRING cluster. Experiments were carried out as three independent replicates.

**Figure 7 cells-11-02673-f007:**
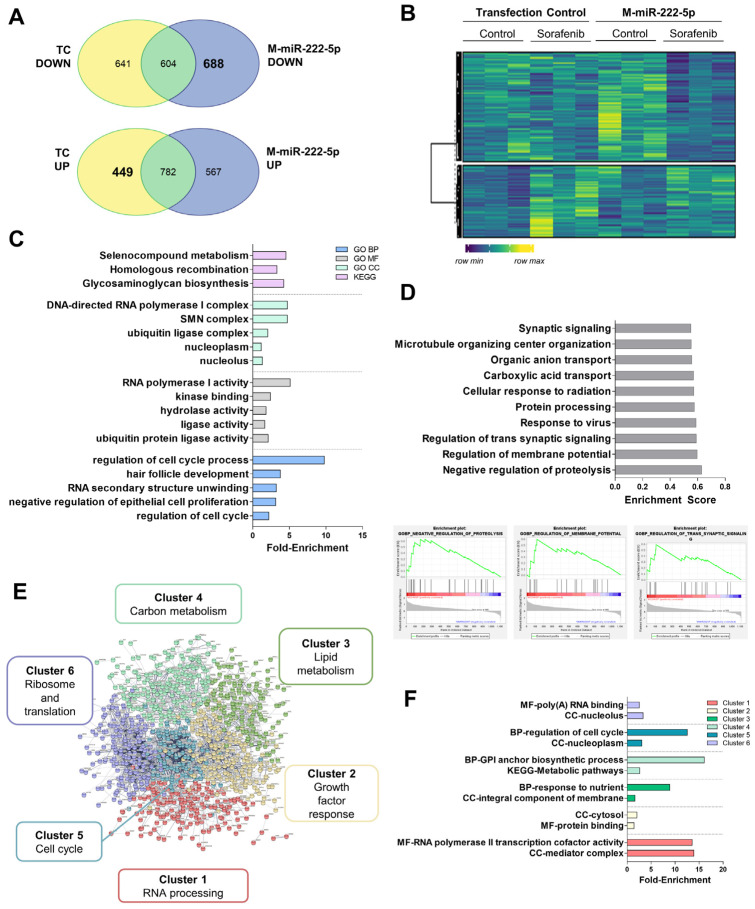
RNA-sequencing data of miR-222-5p overexpression in HepG2 cells showed altered cell cycle and metabolic control. (**A**) Venn diagrams showing mRNAs regulated upon miR-222-5p mimics (M-miR-222-5p) compared to the transfection control (TC). (**B**) Heatmap of genes regulated by miR-222-5p. (**C**) Fold-enrichment values of the top five most significantly enriched GO BP, MF, CC, and KEGG pathways. (**D**) Top 10 significant hallmarks of GSEA downregulated after M-miR-222-5p. Plots were provided for the three most enriched terms. (**E**) Diagram showing clustering of the STRING analysis. (**F**) Fold-enrichment values of the top two most significantly enriched terms either in the GO BP, MF, CC, or KEGG pathways in each STRING cluster. Experiments were carried out as three independent replicates.

**Figure 8 cells-11-02673-f008:**
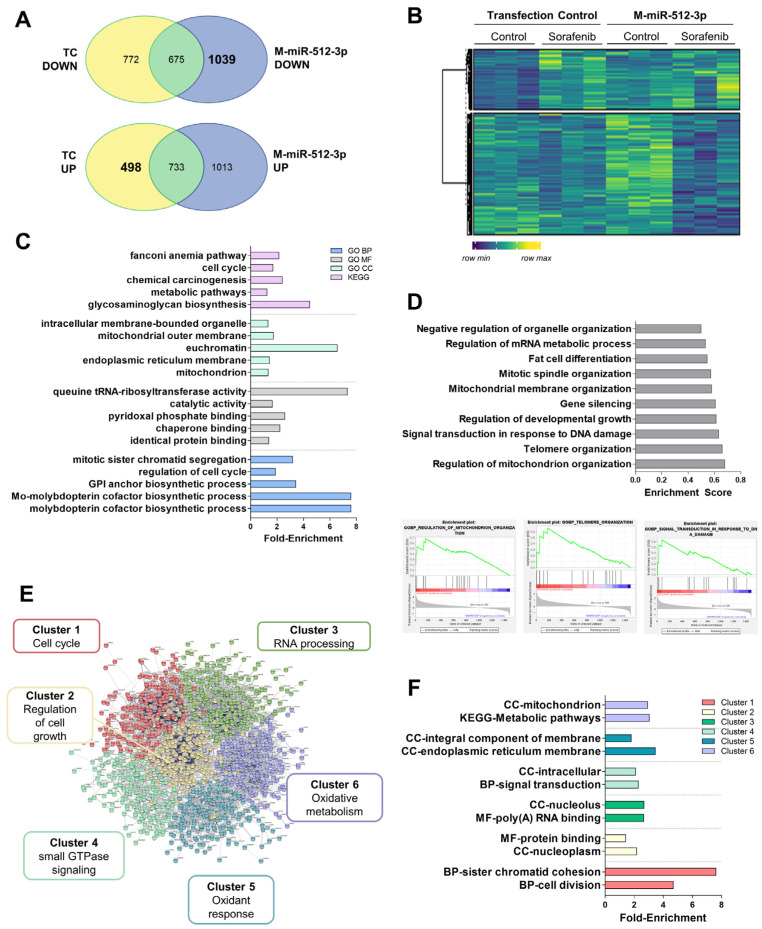
RNA-sequencing data of miR-512-3p overexpression in the HepG2 cells showed reduced oxidative metabolism. (**A**) Venn diagrams showing mRNAs regulated upon miR-512-3p mimics (M-miR-512-3p) compared to the transfection control (TC). (**B**) Heatmap of genes regulated by miR-512-3p. (**C**) Fold-enrichment values of the top five most significantly enriched GO BP, MF, CC, and KEGG pathways. (**D**) Top 10 significant hallmarks of GSEA downregulated after M-miR-222-5p. Plots were provided for the three most enriched terms. (**E**) Diagram showing the clustering of STRING analysis. (**F**) Fold-enrichment values of the top two most significantly enriched terms either in the GO BP, MF, CC, or KEGG pathways in each STRING cluster. Experiments were carried out as three independent replicates.

**Table 1 cells-11-02673-t001:** Summary of the functional studies of miRNAs in cell proliferation, apoptosis, migration, and invasiveness.

miRNA	Proliferation	Apoptosis	Migration	Invasiveness
miR-27a-3p	Not affected	Not affected	Reduction *p* < 0.0001	Reduction *p* < 0.0001
miR-122-5p	Not affected	Reduction *p* < 0.0001	Not affected	Not tested
miR-148b-3p	Increase *p* < 0.0001	Not affected	Not affected	Increase *p* < 0.0001
miR-193b-3p	Not affected	Not affected	Reduction *p* < 0.001	Increase *p* < 0.0001
miR-194-5p	Not affected	Not affected	Increase *p* < 0.0001	Increase *p* < 0.0001
miR-200c-3p	Not affected	Not affected	Reduction *p* < 0.01	Reduction *p* < 0.0001
miR-222-5p	Not affected	Increase *p* < 0.01	Not affected	Increase *p* < 0.0001
miR-375	Not affected	Not affected	Not affected	Not tested
miR-505-5p	Not affected	Reduction *p* < 0.0001	Not affected	Not tested
miR-512-3p	Increase *p* < 0.0001	Not affected	Increase *p* < 0.05	Not tested
miR-551a	Not affected	Not affected	Increase *p* < 0.0001	Reduction *p* < 0.0001

## Data Availability

All data are available in the main text or the Appendix A. The miRNA microarray data have been deposited in the Gene Expression Omnibus with accession code GSE201696 (series GSE201697). The mRNA sequencing experiments have been deposited in the Gene Expression Omnibus with accession code GSE201695 (series GSE201697).

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
