# Peer review of "miR-200c-3p, miR-222-5p, and miR-512-3p Constitute a Biomarker Signature of Sorafenib Effectiveness in Advanced Hepatocellular Carcinoma"

_cells, 2022, doi:10.3390/cells11172673_

Round 1

Reviewer 1 Report

The manuscript “miR-200c-3p, miR-222-5p and miR-512-3p constitute a biomarker signature of Sorafenib effectiveness in advanced hepatocellular carcinoma” contributes to the outgrowing field of miRNA studies with a complex report on a panel of miRNAs important as biomarkers for HCC. The manuscript is well written, there are many techniques used for the experiments, but because the authors analyse many RNAs species some of the conclusion need to be better sustained. My recommendations are as follows:

Major comments:

1. In sub-chapter 3.3, I suggest that the authors provide the figures and statistics for migration, invasiveness, and proliferation, otherwise it is difficult to interpret the data presented in Table 1.

2. The authors claim that that miR-27a-3p, miR-193b-3p, miR-194-5p, miR-200c- 5823p, and miR-505-5p are related to tumor reduction. This assumption comes from the experiments presented in Figure 2D. However, it appears that these miRs are rather related to sorafenib treatment. The data are correlative and are not a proof of interaction. More experiments are needed (for instance, injection of cells overexpressing these miR) to assert the direct relationships between miRs and tumorigenesis.

3. In the discussion section the authors, related to miRNA profiling, imply that Bioinformatic analysis confirmed their relationship with tumor initiation and progression. Please describe in the result section which are the genes important in these processes.

Minor comments:

1.       The affirmation that sorafenib increases EV size is misleading. I would say that sorafenib induces a change in EVs population with an increased number of larger EVs.

2. Figure 3 D. It is not clear what the column headers (a, b, f, g, I, etc.) mean.

Author Response

Thank you very much for your suggestions. We answered point-by-point all your queries. Lines are provided according to the revised version using the “Track Changes” function.

Reviewer 1

The manuscript “miR-200c-3p, miR-222-5p and miR-512-3p constitute a biomarker signature of Sorafenib effectiveness in advanced hepatocellular carcinoma” contributes to the outgrowing field of miRNA studies with a complex report on a panel of miRNAs important as biomarkers for HCC. The manuscript is well written, there are many techniques used for the experiments, but because the authors analyse many RNAs species some of the conclusion need to be better sustained. My recommendations are as follows:

Major comments:

1.In sub-chapter 3.3, I suggest that the authors provide the figures and statistics for migration, invasiveness, and proliferation, otherwise it is difficult to interpret the data presented in Table 1.

We provided the table in order to give summarized information easy to follow. We have included in the revised version a new figure (new figure 3) with the graphs and statistic for miRNA functional studies after Sorafenib treatment, and also maintained the Table 1, because we think that it is completely necessary to fully understand the role of the miRNA per se. Results from figure 3 are now in lines 398-410.

2.The authors claim that that miR-27a-3p, miR-193b-3p, miR-194-5p, miR-200c- 5823p, and miR-505-5p are related to tumor reduction. This assumption comes from the experiments presented in Figure 2D. However, it appears that these miRs are rather related to sorafenib treatment. The data are correlative and are not a proof of interaction. More experiments are needed (for instance, injection of cells overexpressing these miR) to assert the direct relationships between miRs and tumorigenesis.

Effectively, miR-27a-3p, miR-193b-3p, miR-194-5p, miR-200c-3p, and miR-505-5p are related to tumor reduction induced by Sorafenib treatment. Actually, as you point out, we cannot make correlations or conclusions about tumorigenesis with our data. We did not pretend to show that this miRNA profile was the cause of tumor reduction. In that case, we would need other type of mice models to address this question. However, this is not the aim of the study.

In this new version, we have explained more clearly that this miRNA profile is related to Sorafenib treatment to avoid misunderstanding (lines 383-392).

3.In the discussion section the authors, related to miRNA profiling, imply that Bioinformatic analysis confirmed their relationship with tumor initiation and progression. Please describe in the result section which are the genes important in these processes.

At submission we tried to make the text concise to provide the message more clearly, but we also believe that it is relevant to describe which genes from the transcriptomic profile are related to treatment response and how. This comment has been addressed in the Results section, and also, supplementary tables S7, S10 and S13 contain a full list of all the genes dysregulated by miRNAs.

For miR-200c-3p inhibition we included SLC7A11, ATP1B1, MT1B, MT1M, MT1A, PIM3, HSPA1B and NFE2L2 genes (lines 597-508) in the Results section. Their impact in tumor progression has been discussed in lines 622-646 in the Discussion section.

For miR-222-5p mimic we included MT-TF, MT-CO3, FAS, SOX9, TGFB1, AXL, IRS1, SRC, ABCG8, SERPINC1, ACSL4, HMGCR, PLA2G15, FADS3, FADS2, ELOVL6, FH, ALDOC, ENO3, UGP2, RB1 and CCNB1 genes (lines 519-534) in the Results section. Their impact in tumor progression has been discussed in lines 647-673 in the Discussion section.

For miR-512-3p mimic we included SOS2, ARHGEFs, RAB3D, RAP2C, ATG16L1, ATG14, WIPI1, MAOB, ADH4, ADH5, GPX3, GSR, APOM, HP, MSTG1-3, MRPLs, MRPSs, NDUFAF1 genes (lines 545-558) in the Results section. Their impact in tumor progression has been discussed in lines 674-697 in the Discussion section.

Also, we included two examples of common regulated genes by miR-200c-3p inhibition, and miR-222-5p and miR-512-3p mimic, such as SNAI1 or CADH6 genes (lines 559-561, 697-700).

Minor comments:

1.The affirmation that sorafenib increases EV size is misleading. I would say that sorafenib induces a change in EVs population with an increased number of larger EVs.

Thank you very much for this appreciation. This has been corrected in the Abstract, and in the subchapter 3.4 (lines 439-440).

  1. Figure 3 D. It is not clear what the column headers (a, b, f, g, I, etc.) mean.

We included a description of these headers in the figure legend (now Figure 4, lines 464-465). They correspond to multiple comparison statistics.

Reviewer 2 Report

The identification of miRNA signatures in liquid biopsies associated with sorafenib response has important clinical implications. The authors profiled miRNAs in hepatoblastoma HepG2 cells and tested them in animal models, extracellular vesicles (EVs), and plasma from HCC patients. Sorafenib was found to alter the expression of 11 miRNAs in HepG2 cells. Importantly, miR-200c-3p, miR-222-5p and miR-512-3p were found to have prognostic value and contribute to treatment response. This manuscript is well organized. The following minor concerns can improve the manuscript.

1. The progress and importance of miRNAs in tumor prognosis should be elaborated in Intruduction.

2. Line 96 shows the dose of Sorafenib is 200 mg/kg, please verify.

3. The method of some key indicators should be listed in detail, and cannot simply be stated as following the kit method. For example, Caspase-3/7 activity test.

4. The miRNA expression analysis in primary human hepatocytes and hepatoma cells should have been reported in the literature. Therefore, the authors state that their results should be compared, analyzed, and discussed with the literature that has been reported.

Author Response

Thank you very much for your suggestions. We answered point-by-point all your queries. Lines are provided according to the revised version using the “Track Changes” function 

Reviewer 2

The identification of miRNA signatures in liquid biopsies associated with sorafenib response has important clinical implications. The authors profiled miRNAs in hepatoblastoma HepG2 cells and tested them in animal models, extracellular vesicles (EVs), and plasma from HCC patients. Sorafenib was found to alter the expression of 11 miRNAs in HepG2 cells. Importantly, miR-200c-3p, miR-222-5p and miR-512-3p were found to have prognostic value and contribute to treatment response. This manuscript is well organized. The following minor concerns can improve the manuscript.

1.The progress and importance of miRNAs in tumor prognosis should be elaborated in Intruduction.

We included more information about the role of miRNAs in prognosis in lines 62-79. We tried to explain that, although, there are lots of papers that describe the prognosis of HCC according to miRNA expression in tumor samples, while some others provide mechanistic knowledge, more experiments in validation cohorts of patients with HCC are needed to validate data. Also, we had previous information about the usage of miRNAs as biomarkers of Sorafenib response in the Discussion section (lines 616-628). Although informative, these studies had no robust enough statistics to become potential biomarkers in the clinical setting.

2.Line 96 shows the dose of Sorafenib is 200 mg/kg, please verify.

We verified that the dose for subcutaneous mice models was 200 mg/kg.

3.The method of some key indicators should be listed in detail, and cannot simply be stated as following the kit method. For example, Caspase-3/7 activity test.

We detailed several procedures, including HepG2 cell culture (new line 91-100), transfections with mimics and inhibitors (lines 184-192), caspase-3/7 activity and proliferation assays (lines 194-205), as well as EVs isolation (lines 222-233).

4.The miRNA expression analysis in primary human hepatocytes and hepatoma cells should have been reported in the literature. Therefore, the authors state that their results should be compared, analyzed, and discussed with the literature that has been reported.

This aspect has been discussed in lines 588-609. In our manuscript, we compared the expression profile of HepG2 cells to that of primary hepatocytes in order to identify miRNAs related with hepatocarcinogenesis. In fact, we highlighted several miRNAs traditionally involved in this process. Regarding primary human hepatocytes, we included several studies that identify miRNA biomarkers in liver disease. Whereas most of studies rely on hepatocyte cell lines or murine hepatocytes, few studies address this question in human samples, due to the difficulties in sample acquisition.

Reviewer 3 Report

In this manuscript, the authors presented that sorafenib regulates specific miRNA signature in which miR-200c-3p, miR-222-5p and miR-512-5p have prognostic value and contribute to treatment response. It might be useful if these miRNAs play pivotal roles in treatment for advanced HCC.

Major

1.     Many experiments have been done but detailed mechanisms were not clearly shown.  How do Sorafenib change the expression of specific miRNA? If the expression of each miRNA restores to the basal level (before treatment with sorafenib), it becomes effective again? How does specific miRNA cause resistant to sorafenib?

2.    The authors also assessed the trafficking of miRNAs in released EVs by HepG2, but we don’t find the relationships between resistance to sorafenib and these results.

3.    The title is “specific miRNA constitutes a biomarker”, so validation cohort is needed to prove this biomarker is effective.

Minor

1.    Misspelling is in line 586, p17. 512-5p?  3p is correct?

Author Response

Thank you very much for your suggestions. We answered point-by-point all your queries. Lines are provided according to the revised version using the “Track Changes” function 

Reviewer 3

In this manuscript, the authors presented that sorafenib regulates specific miRNA signature in which miR-200c-3p, miR-222-5p and miR-512-5p have prognostic value and contribute to treatment response. It might be useful if these miRNAs play pivotal roles in treatment for advanced HCC.

Major comments:

1.Many experiments have been done but detailed mechanisms were not clearly shown. How do Sorafenib change the expression of specific miRNA? If the expression of each miRNA restores to the basal level (before treatment with sorafenib), it becomes effective again? How does specific miRNA cause resistant to sorafenib?

Thank you very much for your suggestions. However, the approach of this study was different. We profiled miRNA biomarkers of Sorafenib treatment response in HepG2 cells, we validated them in vitro, in vivo and in plasma of patients. Furthermore, we identified a transcriptomic profile regulated by selected miRNAs. Although engaging, our goal was not to define molecular mechanisms governing miRNA expression, but to identify liquid biopsy biomarkers that are urgently needed in the clinical setting.

Regarding the second question, we used mimics and inhibitors to increase or decrease the presence of down-regulated or up-regulated miRNAs. That is, we restored somehow the expression to baseline levels, as you suggested. In the presence of these regulators, we could see that parameters related to Sorafenib effectiveness were altered, including proliferation, cell death, migration or invasion. In fact, we tested if doing so, Sorafenib was more or less effective (Table 1, Figure 3 (new)).

Eventually, we did not profoundly explored mechanisms of resistance to Sorafenib. Instead, we worked with HepG2 hepatoblastoma cell line that is considered to be a good responder of Sorafenib. As a consequence, our selected miRNAs constitute biomarkers of response to Sorafenib, but not of treatment resistance. Nonetheless, we consider this aspect to be very relevant, so we discussed Sorafenib resistance related to the findings obtained in the transcriptomic profiles in the revised manuscript (lines 597-508, 519-534, 545-558, 622-646, 647-673, 674-697). For instance, in the case of miR-200c-3p, we observed that its inhibitor induced the expression of NFE2L2 and MTs. Nrf2 down-regulation seems to be important during cell death induced by Sorafenib. Also, it is very important during ferroptosis induction, and promotes the expression of MTs. These MTs could also be involved in reduced ferroptosis, leading to reduced cell death and constituting a mechanism of resistance.

2.The authors also assessed the trafficking of miRNAs in released EVs by HepG2, but we don’t find the relationships between resistance to sorafenib and these results.

Although exosomal content has been related with treatment resistance in various cancer types, we did not explore molecular mechanisms of Sorafenib resistance, as we are working with models of treatment response, and the approaches are not fully compatible in our study. We observed that response to Sorafenib was related to a shift in the release of EVs in terms of quantity, size and content. Sorafenib induced the release of apoptotic bodies and decreased the release of exosomes, while increasing the expression of a miRNA pattern in EVs (lines 423-445).

3.The title is “specific miRNA constitutes a biomarker”, so validation cohort is needed to prove this biomarker is effective.

Our studies in plasma of patients treated with Sorafenib were done in two different cohorts. A cohort from Hospital University “Virgen del Rocío” (n=36) (June 2015-Au-86 gust 2021, median follow-up of 10.4 months) was used as study population and another 87 cohort of patients from BCLC was used as the validation cohort (n=81) (March 2008-July 88 2011, median follow-up of 11.4 months) (Materials and Methods, line 100-109; Table S1, Figure 5). Also, to reinforce the quality of the data obtained, we performed time-dependent Cox regression models.

Minor comments:

1.Misspelling is in line 586, p17. 512-5p? 3p is correct?

Thank your very much for noticing. It has been corrected to miR-512-3p in line 715.

Round 2

Reviewer 1 Report

The Authors successfully clarified all my comments and modified their manuscript accordingly.  Therefore, I recommend the manuscript for publication.

Reviewer 3 Report

The manuscript was improved and becomes easy to understand.